# Beyond Templates: Dynamic Adaptation of Reasoning Demonstrations via Feasibility-Aware Exploration

## Abstract

Large language models (LLMs) have shown remarkable reasoning capabilities, yet aligning such abilities to small language models (SLMs) remains a challenge due to distributional mismatches and limited model capacity. Existing reasoning datasets, typically designed for powerful LLMs, often lead to degraded performance when directly applied to weaker models. In this work, we introduce Dynamic Adaptation of Reasoning Trajectories (DART), a novel data adaptation framework that bridges the capability gap between expert reasoning trajectories and diverse SLMs. Instead of uniformly imitating expert steps, DART employs a *selective imitation strategy* guided by step-wise adaptability estimation via solution simulation. When expert steps surpass the student's capacity—signaled by an *imitation gap*—the student autonomously explores alternative reasoning paths, constrained by outcome consistency. We validate DART across multiple reasoning benchmarks and model scales, demonstrating that it significantly improves generalization and data efficiency over static fine-tuning. Our method enhances supervision quality by aligning training signals with the student's reasoning capabilities, offering a scalable solution for reasoning alignment in resource-constrained models.

## 1 Introduction

Large language models (LLMs) have recently achieved remarkable performance in complex reasoning tasks such as mathematics and programming (OpenAI, 2024; Shao et al., 2024). A key insight from recent work (Zhou et al., 2024; Yue et al., 2024; Ye et al., 2025) is that small, high-quality instruction datasets are surprisingly effective at eliciting sophisticated reasoning abilities in large models. This discovery challenges traditional beliefs (Li et al., 2024; Yu et al., 2024) that complex cognitive skills necessarily require massive supervised fine-tuning, opening promising avenues for data-efficient model alignment.

Despite the remarkable effectiveness of small, high-quality instruction datasets in eliciting sophisticated reasoning, mainstream approaches (Zhou et al., 2024; Ye et al., 2025; Muennighoff et al., 2025) remain reliant on **static, pre-collected** reasoning datasets. While effective in controlled environments, these datasets struggle to generalize across heterogeneous pretraining distributions, particularly for small language models (SLMs) with diverse training data and limited reasoning capabilities (Xu et al., 2024; Yeo et al., 2025). Disparities in model scale, reasoning proficiency, and training history exacerbate distributional mismatches, significantly hindering the activation of reasoning skills.

To address these challenges, we introduce **Dynamic Adaptation of Reasoning Trajectories (DART)**, a novel data adaptation framework designed to bridge the distribution gap between static reasoning datasets and diverse SLMs. Instead of enforcing uniform imitation of expert demonstrations, DART introduces a *selective imitation* strategy guided by *imitation feasibility estimate*. For each step provided by the expert, DART dynamically assesses the likelihood that the student model can successfully complete the reasoning process conditioned on adopting that step. When imitation is deemed infeasible, the student autonomously explores alternative trajectories while maintaining the consistency of the outcome with the objective of the original task. This approach enables DART to flexibly adapt high-quality reasoning datasets to heterogeneous model populations, significantly improving reasoning elicitation under distribution shift.

In summary, our contributions are as follows.

- We identify the critical limitations of applying static curated reasoning datasets to diverse small language models and propose **DART**, a novel framework for adapted reasoning data guided by imitation feasibility.
- We introduce a Monte Carlo simulation-based method to estimate the feasibility of imitation per step, allowing selective supervision tailored to the student model capabilities.
- We develop an autonomous exploration mechanism that allows models to recover from infeasible supervision points, generating outcome-consistent alternative reasoning paths.

- Through extensive experiments across different model scales and benchmarks, we demonstrate that DART substantially improves reasoning performance over static fine-tuning, achieving superior data efficiency and generalization.

## 2 PRELIMINARIES AND LIMITATIONS OF SUPERVISED IMITATION ON EXPERT TRAJECTORIES

### 2.1 PROBLEM DEFINITION: REASONING CAPABILITY ELICITATION VIA MINIMAL DEMONSTRATIONS

We define the reasoning elicitation problem in the context of large language models (LLMs) with latent pre-trained knowledge. Let $\mathcal{Q}$ denote the space of reasoning problems, $\mathcal{A}$ the space of answers, and $\mathcal{R}$ the space of reasoning chains, where each $r \in \mathcal{R}$ is a sequence of logical steps $r = \{s_1, s_2, \ldots, s_n\}$.

The goal is to learn a reasoning function:

$$f : \mathcal{Q} \to \mathcal{R} \times \mathcal{A} \tag{1}$$

so that, given a question $q \in \mathcal{Q}$, the model generates a logically valid reasoning chain $r \in \mathcal{R}$ and a verifiable final answer $a \in \mathcal{A}$.

Prior work (e.g., (Ye et al., 2025), (Muennighoff et al., 2025)) suggests that reasoning competence in large language models (LLMs) can be elicited not by scale alone, but a small set of carefully crafted demonstrations that expose the underlying cognitive structure of reasoning. This paradigm assumes that latent reasoning skills embedded within pretrained models can be activated through appropriately designed prompts in the form of explicit multi-step exemplars.

Let $\mathcal{D} = \{(q_i, r_i, a_i)\}_{i=1}^{N}$ represent a compact yet high-quality dataset ($N \ll |\mathcal{Q}|$), where each tuple contains a question $q_i$, a structured reasoning chain $r_i$, and its corresponding answer $a_i$. Each $r_i$ serves as a **cognitive template**—an interpretable, step-wise reasoning demonstration designed to guide the model through logical steps with intermediate verification. Instead of introducing new knowledge, these templates activate the model's latent reasoning capabilities by leveraging structured prompting (Wei et al., 2022; Zhou et al., 2024; Ye et al., 2025).

### 2.2 LIMITATIONS OF SUPERVISED IMITATION ON EXPERT DEMONSTRATIONS

Despite its pedagogical appeal, supervised imitation over expert demonstrations exhibits critical limitations when applied to LLMs with diverse capacity levels.

This paradigm (Wei et al., 2022; Ye et al., 2025) assumes that the model possesses sufficient latent competence to internalize and reproduce the reasoning trajectory in each template. In practice, this assumption frequently fails. A template $r_i$ may (i) over-challenge the model by invoking reasoning procedures not encoded in its weights, or (ii) misalign with the model's inductive biases, causing representational mismatch. We define a reasoning failure event $\mathcal{F}$ as the inability of the model to emulate the intended behavior given an input-template pair:

$$\mathcal{F}(f; q, r, a) = \mathbb{I}[f(q) \not\approx (r, a)] \tag{2}$$

where $\mathbb{I}[\cdot]$ is the indicator function. Such failures may arise from superficial imitation, incomplete reasoning chains, or insufficient justification for the final answer.

Compounding this challenge is the substantial cost associated with constructing template datasets $\mathcal{D}$ that satisfy the Cognitive Template Demonstration criterion. Such templates demand meticulous logical decomposition, intermediate verification, and fine-grained pedagogical design. Furthermore, a template crafted for a specific model often fails to generalize to others due to differences in scale, pretraining corpus, or architectural inductive biases, resulting in distributional shifts. As highlighted in prior work on imitation learning (Pomerleau, 1991; Ross et al., 2011), relying on static datasets for training can lead to a distribution mismatch between the output sequences encountered during training and those generated auto-regressively by the student at inference time, undermining generalization and robustness.

**The Need for Imitation Feasibility-Aware Adaptation.**   These limitations highlight the inadequacy of static demonstrations in addressing the diversity of model behaviors. We argue for a dynamic grounding mechanism that aligns template presentation with the target model's internal capacity and abstraction level. Rather than treating $\mathcal{D}$ as fixed input, the elicitation process should adaptively align the demonstrated reasoning path with the model's own preferred or accessible inference trajectories, potentially reformulating how the reasoning unfolds to match internal representations. This motivates our central question:

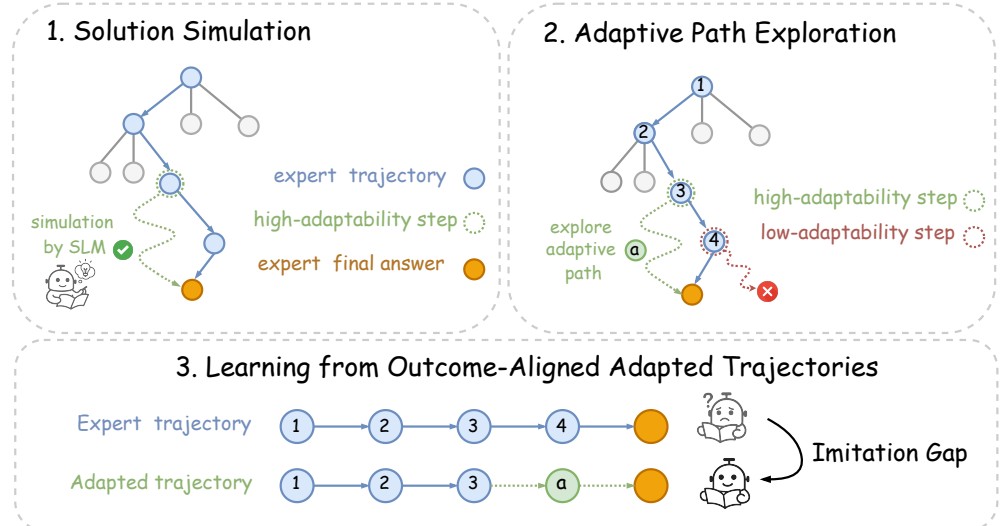

Figure 1: Overview of the DART framework.

*Can we design a dynamic adaptation mechanism that reliably anchors cognitive templates in model-specific latent space, enabling scalable and robust reasoning?*

In the following section, we instantiate this motivation via our proposed framework — **Dynamic Adaptation of Reasoning Trajectories (DART)**.

## 3 METHODOLOGY

In this section, we propose **Dynamic Adaptation of Reasoning Trajectories (DART)**, a capability-aware adaptation framework designed to align expert-level reasoning data with the capacity of small language models (SLMs). Instead of statically mimicking expert trajectories from the elicitation template set, DART introduces a selective imitation mechanism that dynamically adapts supervision signals based on the model's reasoning proficiency. The framework comprises three key components: (1) step-wise adaptability estimation via solution simulation (Section 3.1), (2) imitation gap detection and adaptive path exploration (Section 3.2), and (3) learning from outcome-aligned adapted trajectories (Section 3.3). Figure 1 provides an overview of the pipeline,while Algorithm 1 formalizes the procedure, detailing how these components are operationalized to generate and train on adapted trajectories.

### 3.1 STEP-WISE ADAPTABILITY ESTIMATION VIA SOLUTION SIMULATION

To determine whether a given expert step is suitable for imitation, we introduce the concept of **adaptability**: the likelihood that a student model can reach the correct answer when conditioned on that step. This evaluation is conducted via solution simulation—akin to Monte Carlo Tree Search (Kocsis & Szepesvári, 2006; Silver et al., 2016; Świechowski et al., 2023)—by rolling out multiple completions from partially constructed trajectories that incorporate the candidate step.

Let $s_{<t} = \{s_0, s_1, \ldots, s_{t-1}\}$ be the prefix of expert steps, and $s_t$ the candidate step under evaluation. The adaptability score $f_t$ is computed as:

$$f_t = Q(s_{<t}, s_t) = \frac{1}{N_{\text{sim}}} \sum_{i=1}^{N_{\text{sim}}} \mathbb{I}(a_i^{\text{final}} = a^*) \tag{3}$$

where $N_{\text{sim}}$ denotes the total number of *rollouts* performed for each candidate step $s_t$, with each rollout simulating a complete reasoning trajectory conditioned on the prefix $s_{<t}$ and the adoption of step $s_t$.

Empirically observed patterns (see Section 5.1) suggest that adaptability tends to rise in the early stages of expert trajectories, but drops sharply beyond a certain point. This non-monotonic behavior motivates our definition of the

---

**Algorithm 1** DART: Dynamic Adaptation of Reasoning Trajectories

---

**Require:** Expert trajectory $\tau_{\text{expert}} = \{s_0, s_1, \ldots, s_T\}$, student model $\pi_{\text{student}}$, ground-truth answer $a^*$, adaptation simulation count $N_{\text{sim}}$
1: Initialize prefix $\tau_{\text{prefix}} \leftarrow \emptyset$; adaptability scores $\mathcal{F} \leftarrow []$
2: **for** $t = 0$ to $T$ **do**
3:     Compute adaptability score $f_t \leftarrow Q(s_{<t}, s_t)$         ▷ See Eq. equation 3
4:     Append $f_t$ to $\mathcal{F}$
5: Find $t_{\text{peak}}$ where $f_t$ attains its local maximum;

$$t_{\text{gap}} \leftarrow \min\left\{t > t_{\text{peak}} \mid f_t < f_{t_{\text{peak}}} - \epsilon\right\} \quad \text{where } \epsilon > 0 \text{ defines a significant drop threshold}$$

6: Truncate expert prefix: $\tau_{\text{prefix}} \leftarrow \{s_0, \ldots, s_{t_{\text{gap}}-1}\}$
7: Initialize adapted trajectory: $\tau_{\text{adapt}} \leftarrow \tau_{\text{prefix}}$
8: **while** not terminal **do**
9:     Sample next step $s' \sim \pi_{\text{student}}(\cdot \mid \tau_{\text{adapt}})$
10:    Append $s'$ to $\tau_{\text{adapt}}$
11: **if** $\mathcal{O}_{\text{adapt}}(\tau_{\text{adapt}}) = \mathcal{O}_{\text{expert}}(\tau_{\text{expert}})$ **then**
12:    Retain $\tau_{\text{adapt}}$ for distillation         ▷ See Eq. equation 6
13: **else**
14:    Discard $\tau_{\text{adapt}}$

---

**imitation gap**, a regime in which continued imitation becomes counterproductive due to the increasing complexity of the remaining expert steps.

### 3.2 Adaptive Path Exploration

To avoid overfitting to brittle expert demonstrations, we monitor the *adaptability score* throughout the trajectory and halt imitation once a significant drop is detected (see Equation 3). Motivated by the need to overcome low-adaptability segments that may hinder generalization, DART transitions to autonomous rollout beyond the gap, generating a continuation from the last high-adaptability prefix:

$$\tau_{\text{adapt}} = (s_0, s_1, \ldots, s_{t-1}, s'_t, s'_{t+1}, \ldots, s'_T), \tag{4}$$

where $s'_t, \ldots, s'_T$ are student-generated reasoning steps. Inspired by outcome-based learning strategies (DeepSeek-AI et al., 2025), we do not constrain this trajectory to mimic the expert's form. Instead, we enforce an *outcome consistency* constraint to ensure semantic alignment, as described in Eq. equation 5, as we observe that process supervision (Lightman et al., 2024; Zhang et al., 2025), such as via a Process Reward Model (PRM), often encounters inherent ambiguities and standardization challenges in practice.

$$C(\tau_{\text{adapt}}, \tau_{\text{expert}}) = \begin{cases} 1, & \text{if } \mathcal{O}(\tau_{\text{adapt}}) = \mathcal{O}(\tau_{\text{expert}}), \\ 0, & \text{otherwise.} \end{cases} \tag{5}$$

Here, $C \in \{0, 1\}$ denotes task-level agreement, with $\mathcal{O}(\cdot)$ representing the final answer obtained by executing a reasoning path. Specifically, $\mathcal{O}(\tau_{\text{expert}})$ refers to the outcome of the expert demonstration, while $\mathcal{O}(\tau_{\text{adapt}})$ captures the result of the student's adapted trajectory. The constraint $\mathcal{O}(\tau_{\text{adapt}}) = \mathcal{O}(\tau_{\text{expert}})$ ensures that, although the reasoning paths may differ, their semantic outcomes are equivalent. This outcome consistency criterion allows the student to depart from brittle expert traces while preserving task correctness.

This strategy empowers the student model to develop its own reasoning strategies beyond segments with low adaptability, guided solely by the correctness of the final outcome. By anchoring supervision at the outcome level rather than mimicking intermediate steps, we alleviate the brittleness of process-level imitation. This encourages robust generalization, reduces reliance on ambiguous or inconsistent expert demonstrations, and aligns with the broader goal of enabling flexible yet goal-directed reasoning.

### 3.3 Learning from Outcome-Aligned Adapted Trajectories

To effectively activate the student model's own reasoning ability, we apply a standard cross-entropy loss on the outcome-aligned adapted trajectories generated during autonomous exploration. This training objective encourages the model to reinforce reasoning patterns that are not only aligned with the task goal but also feasible under its own capacity.

Table 1: Main results (%) on LIMO and Math-QwQ-32B across adaptation strategies and model sizes. **Static** overfits to noisy data, while **Adaptation-Full** improves results through exploration and filtering of low-adaptability segments.

| Dataset | Method | GSM8K | GaoKao | Olympiad Bench | College Math | MMLU STEM | Avg. |
|---|---|---|---|---|---|---|---|
| Qwen2.5-0.5B-Instruct | | | | | | | |
| - | No-Tuning | 49.1 | 30.4 | 9.3 | 28.9 | 36.7 | 30.9 |
| Math-QwQ-32B | Static | 39.8 -9.3 | 20.5 -9.9 | 5.9 -3.4 | 17.3 -11.6 | 27.9 -8.8 | 22.3 -8.6 |
| Math-QwQ-32B | **Adaptation-Full** | 49.6 +0.5 | 30.9 +0.5 | 9.3 +0.0 | 27.5 -1.4 | 37.5 +0.8 | 31.0 +0.1 |
| LIMO | Static | 49.6 +0.5 | 26.8 -3.6 | 7.7 -1.6 | 27.3 -1.6 | 32.9 -3.8 | 28.9 -2.0 |
| LIMO | **Adaptation-Full** | 52.2 +3.1 | 32.5 +2.1 | 9.8 +0.5 | 29.1 +0.2 | 37.2 +0.5 | 32.2 +1.3 |
| Qwen2.5-3B-Instruct | | | | | | | |
| - | No-Tuning | 87.0 | 56.6 | 27.3 | 39.9 | 47.6 | 51.7 |
| Math-QwQ-32B | Static | 82.0 -5.0 | 46.2 -10.4 | 20.1 -7.2 | 35.7 -4.2 | 51.3 +3.7 | 47.1 -4.6 |
| Math-QwQ-32B | **Adaptation-Full** | 86.6 -0.4 | 57.9 +1.3 | 29.0 +1.7 | 44.4 +4.5 | 50.6 +3.0 | 53.7 +2.0 |
| LIMO | Static | 85.4 -1.6 | 53.8 -2.8 | 25.2 -2.1 | 41.8 +1.9 | 54.8 +7.2 | 52.2 +0.5 |
| LIMO | **Adaptation-Full** | 87.2 +0.2 | 59.5 +2.9 | 30.4 +3.1 | 43.9 +4.0 | 62.7 +15.1 | 56.7 +5.0 |
| LLaMA-3B-Instruct | | | | | | | |
| - | No-Tuning | 38.4 | 21.3 | 10.7 | 16.0 | 48.0 | 26.9 |
| Math-QwQ-32B | Static | 67.7 +29.3 | 30.4 +9.1 | 10.2 -0.5 | 21.1 +5.1 | 39.2 -8.8 | 33.7 +6.8 |
| Math-QwQ-32B | **Adaptation-Full** | 72.3 +33.9 | 34.0 +12.7 | 10.2 -0.5 | 21.8 +5.8 | 38.3 -9.7 | 35.3 +8.4 |
| LIMO | Static | 28.1 -10.3 | 15.1 -6.2 | 3.6 -7.1 | 11.4 -4.6 | 48.6 +0.6 | 21.4 -5.5 |
| LIMO | **Adaptation-Full** | 47.2 +8.8 | 24.9 +3.6 | 6.4 -4.3 | 18.1 +2.1 | 47.2 -0.8 | 28.8 +1.9 |

Training proceeds by distilling the adapted trajectory $\tau_{\text{adapt}}$ using a standard cross-entropy loss (Kim & Rush, 2016; Bengio et al., 2003):

$$L_{\text{DART}} = -\sum_{t=1}^{T} \mathbb{E}_{(s_{<t}, a_t) \sim \tau_{\text{adapt}}} \left[ \log \pi_{\text{student}}(a_t \mid s_{<t}) \right] \tag{6}$$

Here, $s_{<t} = \{s_0, \ldots, s_{t-1}\}$ denotes the contextual prefix consisting of all prior reasoning steps up to time $t$, and $a_t$ is the corresponding next-step decision. This loss encourages the student model $\pi_{\text{student}}$ to maximize the likelihood of producing $a_t$ when conditioned on its own reasoning history.

By learning from outcome-aligned yet model-compatible trajectories, DART provides high-quality supervision that reflects the student's actual competence. This approach decouples the training signal from rigid trajectory matching, improving both robustness and scalability across models with varying capacity.

## 4 EXPERIMENTS

We evaluate DART across a series of mathematical reasoning benchmarks to assess its effectiveness in adapting expert data to student models of varying capacities.

### 4.1 EXPERIMENTAL SETUP

**Adaptation Datasets.** We conduct adaptation experiments using two datasets. (1) **LIMO** dataset (Ye et al., 2025), a curated set of 817 high-quality math reasoning examples with multi-step CoT demonstrations tailored.We use the official filtered release[1]. (2) The **Math-QwQ-32B** dataset is a synthetic dataset derived from the MATH benchmark

---

[1] https://huggingface.co/GAIR/LIMO

(Hendrycks et al., 2021b), where the Qwen/QwQ-32B-Preview model[2] generates long-form Chain-of-Thought (CoT) solutions for 5,383 problems in the training subset.

**Adaptation Strategies.** We evaluate three adaptation strategies to disentangle the effects of selective imitation and adaptive exploration. (1) *No-Tuning* denotes direct zero-shot evaluation. (2) *Static* reflects standard offline supervised fine-tuning on the full set of expert trajectories, without any filtering or adaptability mechanism. (3) The *Adaptation-Full* strategy represents the complete DART pipeline, integrating imitation gap detection with outcome-consistent student exploration. This approach empowers the model to autonomously explore alternative reasoning paths when expert imitation becomes unreliable. If the model can't find a suitable alternative path, it discards that expert example. We evaluate our method on Qwen2.5-Instruct models at 0.5B[3], 1.5B[4], 3B[5], and LLaMA-3B-Instruct[6] scales, covering a diverse range of SLMs.

**Benchmark Tasks.** We evaluate DART on seven diverse benchmarks encompassing a broad spectrum of mathematical reasoning. These include GSM8K (Cobbe et al., 2021) , covering grade-school to competition-level problems, To assess linguistic and cultural generalization, we incorporate GaoKao 2023 En (Liao et al., 2024), a Chinese national exam benchmark. OlympiadBench (He et al., 2024) features high-difficulty, compositional problems from international math competitions. College Math (Tang et al., 2024) probes undergraduate-level topics in calculus, algebra, and discrete math. MMLU-STEM (Hendrycks et al., 2021a) evaluates STEM-focused reasoning breadth. Overall adaptation is quantified by the arithmetic mean (Avg.) across all benchmarks.

**Training and Model Selection** To ensure the reliability of experimental results, we conducted systematic training and model selection for all models. For both the *Static* and *Adaptation-Full* strategies, we trained models at the 0.5B, 1.5B, and 3B scales for 15 epochs, saving a model checkpoint at the end of each epoch, resulting in 15 checkpoints per model. These checkpoints were evaluated on the validation sets, and the model with the best performance was selected as the final model. The training parameter settings were consistent with LIMO (Ye et al., 2025). All experiments are conducted on the same NVIDIA A100 GPU infrastructure. Additional implementation details, including configurations and setups, are provided in Appendix A.

## 4.2 MAIN RESULTS

Table 1 reports performance across five mathematical reasoning benchmarks, demonstrating the effectiveness of the proposed DART framework in aligning expert reasoning with the capabilities of small language models (SLMs).

**Static Results** Static, which rigidly imitates expert trajectories without adaptation, exhibits clear limitations. On Qwen2.5-0.5B, it decreases accuracy by **8.6 points** on *Math-QwQ-32B* and by **2.0 points** on *LIMO*. On Qwen2.5-3B, Static reduces accuracy on *Math-QwQ-32B* by **4.6 points**. A similar degradation is observed on LLaMA-3B, where accuracy on *LIMO* drops by **5.5 points**. These results are obtained under our careful training and model selection protocol (see Sec. 4.1), ensuring that the observed degradation is not caused by insufficient training but rather reflects the inherent limitations of the Static strategy. These findings indicate that imitating expert demonstrations without adaptation not only constrains small models but can also undermine the performance of larger ones.

**Adaptation-Full Results** Adaptation-Full shows robust improvements over No-Tuning, consistently enhancing performance across datasets and model scales. For example, on the Qwen2.5-0.5B model, Adaptation-Full improves the average accuracy on *LIMO* by +1.3 points, while on the larger Qwen2.5-3B, it yields gains of +2.0 and +5.0 points on *Math-QwQ-32B* and *LIMO*, respectively. The effect is even more pronounced on LLaMA-3B, where Adaptation-Full boosts *Math-QwQ-32B* by +8.4 points and *LIMO* by +1.9 points. On average, Adaptation-Full achieves a **+4.9** point improvement over No-Tuning, demonstrating its effectiveness in aligning reasoning trajectories with model capacity while maintaining stability across different architectures.

## 5 ANALYSIS

We further analyze the internal mechanisms of DART, aiming to understand why selective imitation and autonomous exploration improve reasoning capabilities.

---

[2] https://huggingface.co/Qwen/QwQ-32B-Preview
[3] https://huggingface.co/Qwen/Qwen2.5-0.5B-Instruct
[4] https://huggingface.co/Qwen/Qwen2.5-1.5B-Instruct
[5] https://huggingface.co/Qwen/Qwen2.5-3B-Instruct
[6] https://huggingface.co/meta-llama/LLaMA-3B-Instruct

Figure 2: Step-wise adaptability scores across expert trajectories for Qwen2.5-Instruct student models of varying sizes (0.5B, 1.5B, 3B parameters) under LIMO (top row) and Math-QwQ-32B dataset (bottom row) supervision. The emergence of the **Imitation Gap** is evident: initial steps yield positive adaptation, but continued step-by-step imitation can become harmful.

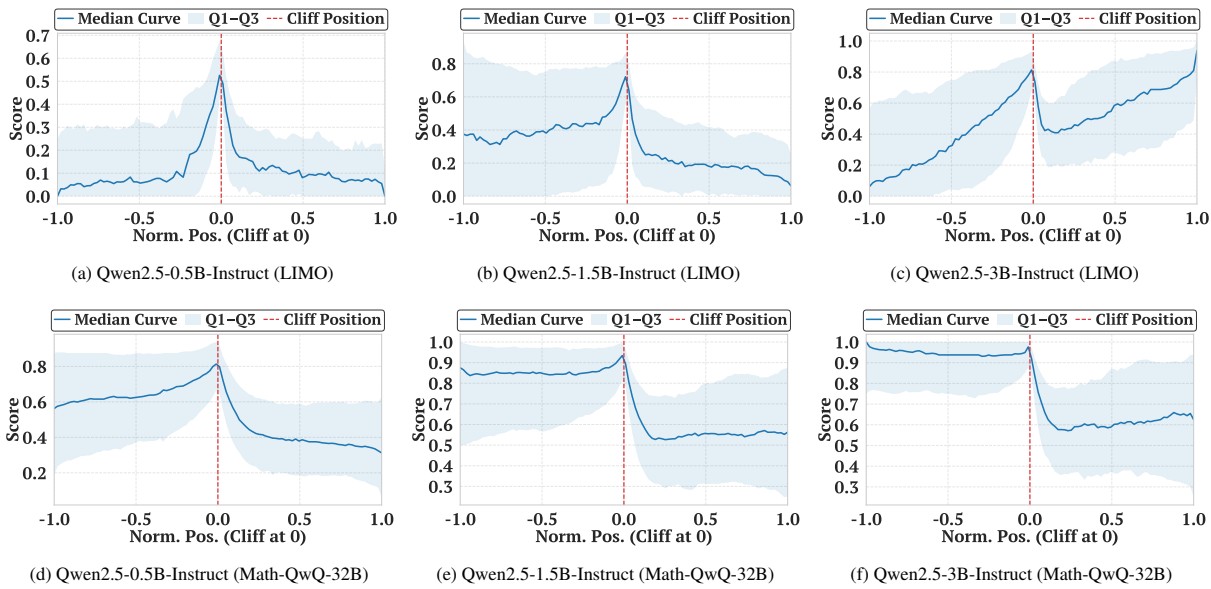

(a) Qwen2.5-0.5B-Instruct (LIMO)  (b) Qwen2.5-1.5B-Instruct (LIMO)  (c) Qwen2.5-3B-Instruct (LIMO)

(d) Qwen2.5-0.5B-Instruct (Math-QwQ-32B)  (e) Qwen2.5-1.5B-Instruct (Math-QwQ-32B)  (f) Qwen2.5-3B-Instruct (Math-QwQ-32B)

## 5.1 STEP-WISE ADAPTABILITY REVEALS THE EMERGENCE OF THE IMITATION GAP

To empirically validate the *imitation gap* hypothesis introduced in Section 3, we estimate the step-wise adaptability scores of Qwen2.5-Instruct student models across three parameter scales (0.5B, 1.5B, 3B) on two reasoning datasets (LIMO and Math-QwQ-32B). Each adaptability score quantifies the model's average probability of reaching the correct final answer when conditioned on imitating a given intermediate step from the expert trajectory. To remove any confounding effect of unequal trajectory lengths, we length-normalize every trace and register the detected cliff at $x = 0$. The curve shows the median adaptability score, and the shaded area the interquartile range (Q1–Q3).

As shown in Figure 2, these curves reveal a consistent behavioral pattern: early in the reasoning path, student models exhibit increasing adaptability as they benefit from following expert steps. However, beyond a certain point, adaptability scores sharply decline—signaling that the student has encountered steps that exceed its reasoning capacity, leading to degraded rollout completions and a collapse in trajectory success.

This non-monotonic pattern reveals the **imitation gap**—a critical region where student models falter due to misalignment between their capabilities and the expert's step distribution. This misalignment arises from distributional discrepancies, where expert trajectories include reasoning patterns outside the student's abstraction space. Consequently, continued imitation in this zone not only fails to benefit learning but actively impairs performance. This phenomenon underscores our central argument: effective reasoning supervision must be dynamically aligned with model-specific capabilities, as realized in our DART framework. To elucidate the imitation gap's impact on adaptability score decline and reasoning performance, we present a case study on the LIMO dataset, pinpointing its onset in a complex reasoning task through Qwen2.5-3B-Instruct's step-wise adaptability scores (see Table 2).

## 5.2 IMPACT OF SEARCH RESTRICTION ON ADAPTATION STRATEGIES

To evaluate the impact of adaptation strategies without autonomous search, we designed two variants: **Adaptation-First** and **Adaptation-Gap**. **Adaptation-First** halts imitation once a feasible solution state is detected, whereas **Adaptation-Gap** monitors adaptability scores and terminates imitation when sharp declines occur, as described in Section 5.1. Table 3 presents the evaluation results on Math-QwQ-32B for 1.5B and 3B Qwen2.5-Instruct models. Both strategies exhibit performance degradation compared to **Adaptation-Full**, highlighting the critical role of autonomous search for recovery in complex reasoning paths. Notably, **Adaptation-Gap** consistently outperforms **Adaptation-First** across all benchmarks, with significant accuracy gains in the average performance (**35.8% vs. 30.3%** for 1.5B and

Table 2: Case Study on the LIMO Dataset: Solutions are generated by Qwen2.5-3B-Instruct. Step-wise adaptability scores are reported for each reasoning step, with steps subsequent to the Imitation Gap (starting from Step 151) highlighted in gray.

| | Scores |
|---|---|
| Q: The vertices of a regular nonagon (9-sided polygon) are to be labeled with the digits 1 through 9 in such a way that the sum of the numbers on every three consecutive vertices is a multiple of 3. Two acceptable arrangements are considered to be indistinguishable if one can be obtained from the other by rotating the nonagon in the plane. Find the number of distinguishable acceptable arrangements. | |
| $\ldots$ | |
| Step 145: Therefore, except in the case of d=1, which would require all digits being same, but which is impossible with unique digits, otherwise, no fix. Therefore, fixed labelings only for identity, which are all our 1296 acceptable labelings. | 0.750 |
| Step 146: Therefore, the number of distinct arrangements is 1296 divided by 9, which is 144. | 0.938 |
| Step 147: So, is 144 the answer? But wait, let me cross-verify. | 0.875 |
| Step 148: Wait, though: in our prior thought process, we had 6 residue assignments, each giving 216 colorings. So 6 × 216 = 1296. | 0.875 |
| Step 149: If we consider the condition that rotating must not create overcounts, then perhaps we had overcounted by factor 9, so we have to divide by 9. 1296 / 9 = 144. | 1.000 |
| Step 150: Alternatively, to think in terms of necklace counting: the number of distinct necklaces with beads labeled 1-9, under rotation, with a constraint that any three consecutive numbers sum to multiple of 3. So, the formula, via Burnside, gives 144. Does that make sense? | 0.938 |
| Step 151: But, another route. Let me think, maybe I have miscalculated for fixed labelings. Wait, notice that the fixed labelings under rotation by k=3 positions? | 0.438 |
| Step 152: Wait, because d = gcd(3,9) = 3, so the cycles decompose into 3 cycles, each of length 3. So, fixed labelings here would have to assign the same digit to each cycle, but since labels must be unique, is it possible? | 0.688 |
| Step 153: Wait, meaning that for each cycle (1,4,7), (2,5,8), (3,6,9); all three digits in cycle (1,4,7) must be the same, similarly for others, which is impossible with unique digits. | 0.938 |

Table 3: Accuracy (%) on the Math-QwQ-32B dataset for Qwen2.5-1.5B-Instruct and Qwen2.5-3B-Instruct models under different adaptation strategies. **Adaptation-First** performs early stopping at feasible solution states, while **Adaptation-Gap** selectively truncates imitation paths based on adaptability declines. **Adaptation-Full** integrates autonomous search, achieving the highest performance across benchmarks. **Bold** values indicate the best results in each group.

| Model | Method | GSM8K | GaoKao | Olympiad Bench | College Math | MMLU STEM | Avg. |
|---|---|---|---|---|---|---|---|
| | Adaptation-First | 41.2 | 33.0 | 10.7 | 24.9 | 41.7 | 30.3 |
| Qwen2.5-1.5B-Instruct | Adaptation-Gap | 60.0 | 34.5 | 13.6 | 29.7 | 41.3 | 35.8 |
| | **Adaptation-Full** | **74.2** | **48.6** | **19.6** | **39.4** | **57.7** | **47.9** |
| | Adaptation-First | 36.7 | 32.7 | 12.9 | 26.1 | 46.8 | 31.0 |
| Qwen2.5-3B-Instruct | Adaptation-Gap | 77.9 | 43.4 | 18.4 | 36.0 | 44.0 | 43.9 |
| | **Adaptation-Full** | **86.6** | **57.9** | **29.0** | **44.4** | **50.6** | **53.7** |

**43.9% vs. 31.0%** for 3B). This improvement stems from its capacity-aware truncation, which effectively filters out low-adaptability segments, preventing error propagation and enhancing stability.

### 5.3 CAPACITY-ALIGNED LEXICAL DYNAMICS UNDER ADAPTATION

To investigate how DART reshapes student model behavior at different scales, we analyze keyword frequency changes between static and adapted dataset. Table 4 lists the top 20 tokens with the largest shifts in the first sentence of each reasoning step for the Qwen2.5-Instruct series 0.5B, 1.5B, and 3B models.

Adaptation reduces exploratory terms like *but*, *wait*, and *alternatively*, while amplifying goal-oriented expressions such as *step*, *solve*, *find*, and *need*. In the 1.5B model, *but* and *wait* drop by 0.36% and 0.20% percentage points, while *find* and *need* rise by 0.13% and 0.14% points. This shift reflects a transition from hesitant exploration to decisive, solution-driven reasoning. These changes reduce uncertainty and digression—traits often seen in expert trajectories

Table 4: Top 20 Keyword Frequency Changes Across Model Sizes

| Keyword | 0.5B (%) | | | 1.5B (%) | | | 3B (%) | | |
|---|---|---|---|---|---|---|---|---|---|
| | Static | Adapted | $\Delta$ | Static | Adapted | $\Delta$ | Static | Adapted | $\Delta$ |
| but | 2.73 | 2.59 | −0.14 | 2.73 | 2.37 | −0.36 | 2.73 | 2.27 | −0.46 |
| alternatively | 0.86 | 0.79 | −0.07 | 0.86 | 0.72 | −0.14 | 0.86 | 0.71 | −0.15 |
| wait | 2.30 | 2.23 | −0.07 | 2.30 | 2.10 | −0.20 | 2.30 | 2.00 | −0.30 |
| therefore | 1.55 | 1.50 | −0.05 | 1.55 | 1.43 | −0.13 | 1.55 | 1.40 | −0.15 |
| check | 0.51 | 0.47 | −0.04 | 0.51 | 0.40 | −0.11 | 0.51 | 0.33 | −0.18 |
| another | 0.29 | 0.26 | −0.03 | 0.29 | 0.20 | −0.09 | 0.29 | 0.17 | −0.12 |
| then | 0.97 | 0.94 | −0.03 | - | - | - | - | - | - |
| pi | 0.11 | 0.09 | −0.02 | - | - | - | - | - | - |
| perhaps | 0.55 | 0.53 | −0.02 | - | - | - | - | - | - |
| length | 0.22 | 0.24 | +0.02 | - | - | - | - | - | - |
| step | 0.27 | 0.30 | +0.02 | 0.27 | 0.37 | +0.10 | 0.27 | 0.41 | +0.14 |
| now | 0.43 | 0.46 | +0.03 | 0.43 | 0.49 | +0.06 | 0.43 | 0.50 | +0.07 |
| first | 0.89 | 0.92 | +0.03 | 0.89 | 0.96 | +0.07 | 0.89 | 0.96 | +0.07 |
| since | 0.80 | 0.83 | +0.03 | - | - | - | - | - | - |
| have | 0.64 | 0.67 | +0.03 | 0.64 | 0.69 | +0.05 | - | - | - |
| let | 1.83 | 1.86 | +0.04 | 1.83 | 1.88 | +0.05 | - | - | - |
| need | 0.54 | 0.58 | +0.04 | 0.54 | 0.68 | +0.14 | 0.54 | 0.69 | +0.16 |
| find | 0.37 | 0.42 | +0.04 | 0.37 | 0.50 | +0.13 | 0.37 | 0.54 | +0.16 |
| newline | - | - | - | 0.00 | 0.05 | +0.05 | - | - | - |
| equation | - | - | - | 0.76 | 0.81 | +0.05 | 0.76 | 0.85 | +0.09 |

but burdensome for smaller models. In static supervision, such expressions appear frequently, straining low-capacity models and widening the *Imitation Gap* (Sec. 3.1), where expert strategies exceed model capabilities.

DART bridges this gap by replacing brittle reasoning paths with model-originated decision traces. This adaptation maintains task objectives while restructuring execution to fit model capacity, leading to stable and efficient reasoning.

## 6 RELATED WORK

**Chain-of-Thought Reasoning** Early work on chain-of-thought reasoning (CoT) (Wei et al., 2022) primarily focused on *short CoT*, where models generate concise reasoning paths to solve problems. Recent advances (Chen et al., 2025) have shifted towards *long CoT prompting*, encouraging more elaborate reasoning chains that enable systematic exploration of multiple paths (*branching*) and backtracking when errors are detected. While techniques like knowledge distillation (Hinton et al., 2015; Luo et al., 2025) and reinforcement learning (Hou et al., 2025) have been used to equip large language models (LLMs) with long CoT capabilities, these efforts remain largely confined to models with substantial parameter sizes. In contrast, our work specifically addresses the unique challenges associated with training smaller-scale models for complex reasoning tasks.

**Data-Efficient Reasoning Elicitation** A related line of work investigates how minimal supervision can elicit latent reasoning abilities in pretrained models (Ye et al., 2025; Muennighoff et al., 2025). These methods rely on a few carefully designed *cognitive templates*, to guide reasoning, but often assume that models possess the necessary prior knowledge. This assumption makes the templates brittle when cognitive demands exceed model capacity. To address this limitation, we propose a feasibility-aware adaptation framework that dynamically adjusts supervision to model ability, enabling robust reasoning across diverse capacity profiles.

## 7 CONCLUSION

We propose Dynamic Adaptation of Reasoning Trajectories (DART), a data adaptation framework designed to improve reasoning elicitation for small language models. By introducing adaptability-based selective imitation and outcome-consistent exploration, our method aims to better align expert demonstrations with model capabilities. Experimental results across several benchmarks show that DART can improve reasoning performance compared to static fine-tuning. We hope this work provides a step toward more flexible and model-aware data alignment strategies for reasoning tasks.

## 8 LIMITATIONS AND FUTURE WORK

Our framework is effective for structured reasoning tasks with verifiable outcomes. However, its extension to open-ended tasks with inherent output uncertainty remains limited, suggesting the need for refined supervision mechanisms and evaluation metrics to ensure outcome consistency.

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

APPENDIX

## A    IMPLEMENTATION DETAILS

In this section, we provide a detailed account of the experimental configurations and setups to ensure the transparency and reproducibility of our research. We introduce the prompt designs used in our experiments. Furthermore, we elaborate on the parameter configurations for simulation experiments and adaptive path exploration. These configurations are designed to balance computational efficiency and response diversity, ensuring the stability and adaptability of the model across various tasks.

### A.1    EXPERIMENT PROMPTS

In our simulation experiments, we employed a structured prompting approach to guide the language model through multi-step reasoning tasks. The primary simulation prompt used in our study is defined as follows:

```
Simulation Prompt

Problem:  data["question"]
Existing reasoning path:  data["answer"]
Guidelines for continuing the reasoning:
1.  Understand the existing path:  Carefully analyze the existing reasoning
path and understand the logic and basis of each step.
2.  Identify the next step:  Based on the last step of the existing path,
determine the possible directions for the next step of reasoning.
3.  Reason step-by-step:  Start from the last step of the existing path and
proceed with the reasoning step-by-step.
4.  Final conclusion:  When the reasoning is complete, put your final
answer within boxed{}.
Continue reasoning step by step, and put your final answer within \boxed{}.
```

This prompt encourages the model to decompose the problem into intermediate steps and to clearly indicate the final answer using LaTeX-style boxed notation. This formatting ensures consistency across outputs and facilitates automated evaluation of results.

In addition to standard simulation prompting, we introduce a dedicated exploration prompt tailored for the adaptive trajectory rollout described in Section 3.2. This prompt is activated once a low-adaptability segment is detected and aims to continue reasoning beyond the imitation gap. It conditions the model on the prefix of high-adaptability reasoning steps and allows for autonomous continuation constrained only by outcome correctness:

```
Exploration Prompt

Problem:  {data["question"]}
Existing reasoning (read-only; cite only key points when anchoring, do NOT
restate the whole text):  {data["answer"]}
[Guidelines (strictly follow)]
1.  Role & Boundaries
- Continue only from the last step of the existing reasoning; do not
restate or rewrite prior content.
- If new symbols/variables are needed, first define their meaning and
domain in one sentence, then use them.
2.  Anchoring & Continuation
- Use one line to anchor the key equation/state of the "last step" (key
points only; do not restate the full text).
- If you can determine the next step number from previous steps, continue
that numbering; if not, do not number--start reasoning directly.
3.  Explore
- Following your own reasoning style and anchored to what has been
established, carry the reasoning forward from here.
Final conclusion:  When the reasoning is complete, put your final answer
within \boxed{}.
```

This exploration prompt encourages the model to develop its own reasoning path from the last trustworthy segment, fostering flexible generalization while maintaining semantic alignment with the expert outcome.

### A.2 Parameter Configuration for Simulation

The simulation procedure in Algorithm 3.1 adopts stochastic decoding to explore alternative reasoning paths beyond expert demonstrations. We sample $N = 4$ candidate continuations per step, corresponding to the adaptation simulation count $N_{\text{sim}}$.

Each trajectory is generated with a maximum length of `MAX_NEW_TOKENS = 4000`. To promote determinism while retaining minimal stochasticity, we set the sampling temperature to `TEMPERATURE = 0.1`. Decoding is performed in batches of `BATCH_SIZE = 32` to enable efficient parallel inference under hardware constraints. These settings ensure stable simulation rollouts with low-variance outputs, suitable for evaluating adaptability under controlled decoding conditions.

Regarding the computational cost of our simulation, we employ SGLang as the inference deployment framework for small-scale models. Given that our approach primarily focuses on adapting reasoning templates to small-scale models, it encounters challenges stemming from distributional mismatches and the limited capacity of small language models (SLMs) compared to large language models (LLMs). Consequently, we are able to deploy our model on a single GPU. To enhance simulation efficiency, we utilize a distributed rollout engine, with multiple SGLang workers managed by an SGLang router to achieve load balancing. Within our code framework, for a 3B model with an estimated four simulations per step, processing the LIMO dataset on a single node equipped with eight A100 GPUs requires approximately six hours.

### A.3 Parameter Configuration for Adaptive Path Exploration

To support the adaptive rollout mechanism described in Section 3.2, we configured the EXPLORE phase with carefully selected hyperparameters to balance computational efficiency and response diversity. The sampling procedure was executed with a candidate beam size of `NUM_SAMPLES = 8`, meaning that at each decision step, eight reasoning continuations were generated for evaluation based on the adaptability score.

We set the maximum generation length to `MAX_NEW_TOKENS = 2000` to allow sufficient space for multi-step reasoning without premature truncation. A temperature of `TEMPERATURE = 0.7` was employed to introduce moderate randomness in token sampling, facilitating the exploration of alternative reasoning paths while retaining coherence.

Batch inference was performed with a `BATCH_SIZE = 64` to utilize GPU resources efficiently during large-scale rollouts. The underlying language model was run using half-precision arithmetic (`DTYPE = float16`), which reduced memory footprint and improved throughput without compromising output quality.

Additionally, the maximum number of concurrent sequences handled by the inference engine (VLLM) was set to `MAX_NUM_SEQS = 512`, enabling high-throughput parallel generation during exploration. These settings ensured scalable, stable, and semantically diverse adaptation rollouts that align with the outcome consistency constraint described in Equation equation 5.

### B Impact of Search Path Quality on Model Performance

To investigate the impact of search quality on model performance, we conducted a comparative experiment (see Table 5). After completing the adaptation path search, we removed paths exhibiting severe repetition phenomena. As illustrated in Figure 3, the proportion of repeated paths during exploration decreases progressively with increasing model parameter size, indicating that improvements in the model's generative capability and contextual memory effectively reduce repetition.

We refer to the results after removing such repeated paths as *Adaptation-Cleaned* and systematically evaluated these against the complete search results without removing repeated paths, denoted as *Adaptation-Raw*. Experimental results demonstrate that filtering out repeated paths leads to significant performance gains, further highlighting the critical role of search path quality in overall model performance.

### C Comparative Analysis of Truncation Methods under Search Constraints

In our previous section (see Section 5.2), we investigate two truncation methods under different search constraints. Specifically, we designed two variants: **Adaptation-First** and **Adaptation-Gap**. The **Adaptation-First** method

Table 5: Comparison of accuracy (%) on the Math-QwQ-32B dataset for 0.5B and 1.5B models under different adaptation strategies. The table contrasts the performance between *Adaptation-Raw* (without removing repeated paths) and *Adaptation-Cleaned* (with repeated paths removed). Columns for MATH and Minerva Math are excluded, and the average is computed over the remaining datasets. **Bold** values indicate the best results.

| Model | Method | GSM8K | GaoKao | Olympiad Bench | College Math | MMLU STEM | Avg. |
|---|---|---|---|---|---|---|---|
| **Math-QwQ-32B Dataset** | | | | | | | |
| Qwen2.5-0.5B-Instruct | Adaptation-Raw | 47.5 | 29.1 | 9.3 | 26.8 | 28.2 | 28.2 |
| | Adaptation-Cleaned | **49.6** | **30.9** | **9.3** | **27.5** | **37.5** | **31.0** |
| Qwen2.5-1.5B-Instruct | Adaptation-Raw | 70.8 | 44.2 | 18.8 | 38.3 | 44.6 | 43.3 |
| | Adaptation-Cleaned | **74.2** | **48.6** | **19.6** | **39.4** | **57.7** | **47.9** |

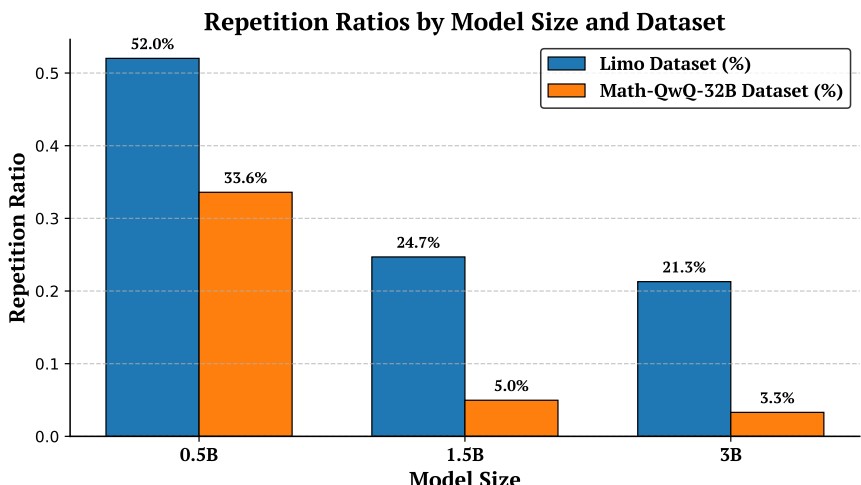

Figure 3: Repetition ratios(%) in search paths across different model sizes and datasets. Smaller models tend to have higher repetition ratios, particularly on the Limo dataset.

halts imitation once a feasible solution state is detected, whereas **Adaptation-Gap** monitors adaptability scores and terminates imitation when sharp declines occur, as detailed in Section 5.1.

We compare the truncation positions of the two methods across different datasets and model sizes. Our analysis indicates that on more challenging datasets, or when the model capacity is limited (e.g., results on the 0.5B models for both datasets), the truncation points identified by **Adaptation-First** and **Adaptation-Gap** are largely consistent. This can be attributed to the complexity of the reasoning cognitive templates in these datasets relative to the model's capabilities: once the model identifies a path leading to a feasible solution, continued imitation often ventures into regions that are difficult to adapt to, typically accompanied by a sharp decline in adaptability scores. Consequently, the truncation positions under both **Adaptation-First** and **Adaptation-Gap** modes are generally aligned.

Conversely, on the Math-Qwen dataset, notable differences in truncation positions emerge. Many models, after reaching the step at which the final answer can be searched, continue to utilize subsequent adaptable path segments. Thus, the **Adaptation-Gap** method is able to detect and leverage a greater number of these usable step fragments, resulting in more substantial performance improvements, as reported in Table 3.

## D  PROOF OF EXISTENCE OF IMITATION GAP

To rigorously establish the existence of the imitation gap in behavioral cloning for reasoning tasks, we model the process as a deterministic Markov Decision Process (MDP) $M = (\mathcal{S}, \mathcal{A}, \mathcal{T}, r, \rho, T)$ (Li & Li), where:

- $\mathcal{S}$: state space of reasoning prefixes including the initial instruction $x$;
- $\mathcal{A}$: action space of reasoning steps $s_t$;

Table 6: Comparison of truncation positions between **Adaptation-First** and **Adaptation-Gap** methods across datasets and model sizes. The relative localization difference represents the absolute difference between the relative truncation positions of these two methods. Higher differences are highlighted with deeper red.

| Dataset | Model Size | First Position | Gap Position | Relative Localization Difference |
|---|---|---|---|---|
| | 0.5B | 0.7901 | 0.7707 | 0.0194 |
| Limo Dataset | 1.5B | 0.6785 | 0.6985 | 0.0200 |
| | 3B | 0.5718 | 0.5983 | 0.0265 |
| | 0.5B | 0.5055 | 0.5244 | 0.0189 |
| Math-QwQ-32B Dataset | 1.5B | 0.2113 | 0.3664 | 0.1551 |
| | 3B | 0.0840 | 0.3239 | 0.2399 |

- $\mathcal{T} : \mathcal{S} \times \mathcal{A} \to \mathcal{S}$: deterministic transition appending $s_t$ to the prefix;
- $r : \mathcal{S} \to \mathbb{R}$: reward function, with $r(s_T) = 1$ if the trajectory yields the correct answer $a^*$, and 0 otherwise;
- $\rho$: initial distribution over instructions ($s_0 \sim \rho$);
- $T$: maximum trajectory length (horizon).

Consider an expert trajectory $\tau_{\text{expert}} = (s_0, s_1, \ldots, s_T)$ generated by a strong policy $\pi_E$, assumed to produce near-optimal steps. The student policy $\pi_S$, trained via behavioral cloning (BC) on expert demonstrations, minimizes the loss $\mathbb{E}_{\tau \sim d^{\pi_E}} \left[ \sum_{t=1}^{T} - \log \pi_S(s_t | s_{<t}) \right]$, where $s_{<t} = (s_0, \ldots, s_{t-1})$ is the prefix, and $d^{\pi_E}$ is the expert state distribution.

Define the Q-value under $\pi_S$ for appending the expert action $s_t$ at prefix $s_{<t}$:

$$f_t = Q^{\pi_S}(s_{<t}, s_t) = \mathbb{E}_{s_{t+1:T} \sim \pi_S(\cdot | s_{\leq t})} \left[ \mathbb{I}(\mathcal{O}(\tau) = a^*) \right],$$

where $\tau = (s_0, \ldots, s_T)$, $\mathcal{O}(\tau)$ extracts the final answer, $a^*$ is the ground truth, $\mathbb{I}$ is the indicator function, and $s_{\leq t} = (s_0, \ldots, s_t)$. Following (Li & Li), we use the sigmoid-transformed Q-value for probability interpretations:

$$f_t^{\sigma} = \sigma(f_t) = \mathbb{P}^{\pi_S}(\mathcal{O}(\tau) = a^* | s_{\leq t}).$$

**Lemma 1** (Existence of Imitation Gap). *There exists a step $t_{gap} \in [1, T]$ such that the sequence of $f_t$ values satisfies $f_1 < f_2 < \cdots < f_{t_{gap}-1}$, followed by a sharp drop $f_{t_{gap}} \ll f_{t_{gap}-1}$.*

*Proof.* The proof is structured in three parts, leveraging Q-value rankings from process reward models (Li & Li) and the impact of distribution mismatch on the student policy.

**Part 1: Pre-gap monotonic increase.** For $t < t_{\text{gap}}$, the prefixes $s_{<t}$ remain aligned with $d^{\pi_E}$, as the student policy $\pi_S$ closely approximates $\pi_E$. Since the expert actions $s_t$ are correct, we apply Lemma 3.3 from (Li & Li): for two correct steps $s_n, s_m$ in a solution $\tau$ with $n < m$, we have:

$$Q^*(s_{<n}, s_n) < Q^*(s_{<m}, s_m).$$

The proof, adapted to our student policy:

$$f_n^{\sigma} - f_m^{\sigma} = \mathcal{P}^{\pi_S}(s_m | s_{<n}) \mathcal{P}^{\pi_S}(\tau | s_{\leq m}) + \mathcal{P}^{\pi_S}(\overline{s_m} | s_{<n}) \mathcal{P}^{\pi_S}(\tau | \overline{s_{\leq m}}) - \mathcal{P}^{\pi_S}(\tau | s_{\leq m}),$$

$$= \mathcal{P}^{\pi_S}(\overline{s_m} | s_{<n}) \left[ \mathcal{P}^{\pi_S}(\tau | \overline{s_{\leq m}}) - \mathcal{P}^{\pi_S}(\tau | s_{\leq m}) \right],$$

where the first equality uses the Q-function definition, and the second uses $\mathcal{P}^{\pi_S}(s_m | s_{<n}) + \mathcal{P}^{\pi_S}(\overline{s_m} | s_{<n}) = 1$. Under Assumption 3.1, $\mathcal{P}^{\pi_S}(\tau | \overline{s_{\leq m}}) - \mathcal{P}^{\pi_S}(\tau | s_{\leq m}) < 0$, since the correct step $s_m$ has a higher probability of leading to a correct outcome. Thus, for $n < m$, $f_n^{\sigma} < f_m^{\sigma}$, implying $f_n < f_m$. Since $\pi_S \approx \pi_E$ for early steps, this holds for all $t < t_{\text{gap}}$, yielding:

$$f_1 < f_2 < \cdots < f_{t_{\text{gap}}-1}.$$

**Part 2: Distribution mismatch and emergence of non-optimal step.** Due to differences in model capacity (e.g., the student being a smaller model), the expert data distribution $d^{\pi_E}$ and the student model distribution $d^{\pi_S}$ are inconsistent. As the number of steps increases, the prefixes $s_{<t}$ grow increasingly complex, becoming likely to fall outside the training distribution of $\pi_S$. Consequently, the state observed by $\pi_S$ at step $t$ diverges from that of $\pi_E$, such that the

expert action $s_t$, optimal under $\pi_E$, is not necessarily optimal under $\pi_S$. This distribution mismatch leads to a critical step $t_{\text{gap}}$ where the expert action $s_{t_{\text{gap}}} = s_E$ is non-optimal for $\pi_S$, as it does not maximize the expected reward under the student's policy:

$$Q^{\pi_S}(s_{<t_{\text{gap}}}, s_E) < \max_{s \in \mathcal{A}} Q^{\pi_S}(s_{<t_{\text{gap}}}, s).$$

This non-optimality arises because the OOD prefix $s_{<t_{\text{gap}}}$ causes $\pi_S$ to misjudge the value of $s_E$, favoring an alternative action that aligns better with its biased distribution, analogous to selecting an incorrect step from a correct prefix.

**Part 3: Sharp drop behavior.** At $t_{\text{gap}}$, appending the non-optimal expert action $s_{t_{\text{gap}}}$ produces an OOD state $s_{\leq t_{\text{gap}}}$, significantly reducing the probability of correct completion. We compare the Q-value of the correct prefix at $t_{\text{gap}} - 1$ to the non-optimal step at $t_{\text{gap}}$. For the correct prefix at $t_{\text{gap}} - 1$, let $s_{t_{\text{gap}}-1}$ be correct, and for the non-optimal step $s_{t_{\text{gap}}}$, we have:

$$f^\sigma_{t_{\text{gap}}-1} - \mathcal{V}^{\pi_S}(x) = \mathcal{P}^{\pi_S}(\overline{s_{t_{\text{gap}}-1}}|x)\left(\mathcal{P}^{\pi_S}(\tau|s_{\leq t_{\text{gap}}-1}) - \mathcal{P}^{\pi_S}(\tau|\overline{s_{\leq t_{\text{gap}}-1}})\right),$$

$$f^\sigma_{t_{\text{gap}}} - \mathcal{V}^{\pi_S}(x) = \mathcal{P}^{\pi_S}(s_{t_{\text{gap}}}|x)\left(\mathcal{P}^{\pi_S}(\tau|\overline{s_{\leq t_{\text{gap}}}}) - \mathcal{P}^{\pi_S}(\tau|s_{\leq t_{\text{gap}}})\right),$$

where $\mathcal{V}^{\pi_S}(x) = \mathbb{P}^{\pi_S}(\mathcal{O}(\tau) = a^*|x)$. Under Assumption 3.1, $\mathcal{P}^{\pi_S}(\tau|s_{\leq t_{\text{gap}}-1}) > \mathcal{P}^{\pi_S}(\tau|\overline{s_{\leq t_{\text{gap}}-1}})$, so the first difference is positive, implying $f^\sigma_{t_{\text{gap}}-1} > \mathcal{V}^{\pi_S}(x)$. For the non-optimal step, $\mathcal{P}^{\pi_S}(\tau|\overline{s_{\leq t_{\text{gap}}}}) < \mathcal{P}^{\pi_S}(\tau|s_{\leq t_{\text{gap}}})$, and since $s_{t_{\text{gap}}}$ is non-optimal due to distribution mismatch, $\mathcal{P}^{\pi_S}(s_{t_{\text{gap}}}|x) \gg \mathcal{P}^{\pi_S}(\overline{s_{t_{\text{gap}}}}|x)$, amplifying the negative difference. Thus:

$$f^\sigma_{t_{\text{gap}}} < \mathcal{V}^{\pi_S}(x) < f^\sigma_{t_{\text{gap}}-1},$$

implying $f_{t_{\text{gap}}} \ll f_{t_{\text{gap}}-1}$, as the non-optimal step's Q-value is significantly lower due to the low probability of recovery from incorrect branches.

The key size relation for the drop is:

$$f^\sigma_{t_{\text{gap}}} - f^\sigma_{t_{\text{gap}}-1} = \mathcal{P}^{\pi_S}(s_{t_{\text{gap}}}|s_{<t_{\text{gap}}})\left[\mathcal{P}^{\pi_S}(\tau|s_{\leq t_{\text{gap}}}) - \mathcal{P}^{\pi_S}(\tau|\overline{s_{\leq t_{\text{gap}}}})\right] + \mathcal{P}^{\pi_S}(\overline{s_{t_{\text{gap}}}}|s_{<t_{\text{gap}}})\left[\mathcal{P}^{\pi_S}(\tau|\overline{s_{\leq t_{\text{gap}}}}) - \mathcal{P}^{\pi_S}(\tau|s_{\leq t_{\text{gap}}})\right] < 0,$$

where the negative term dominates under Assumption 3.1, ensuring $f_{t_{\text{gap}}} \ll f_{t_{\text{gap}}-1}$. $\square$

## E  DESCRIPTION OF LARGE LANGUAGE MODEL USAGE

In the preparation of this manuscript, we leveraged a large language model (LLM), specifically Grok developed by xAI, to facilitate specific aspects of the writing process. The LLM was employed primarily for linguistic refinement, encompassing tasks such as enhancing sentence coherence, improving syntactic clarity, and elevating the overall readability of the text, while preserving the integrity of the scientific content, methodologies, and findings. The rationale for this approach was to optimize the communicative efficacy of the manuscript, ensuring that intricate technical concepts are articulated with precision and accessibility for a diverse academic readership. All outputs generated by the LLM were subjected to rigorous scrutiny, validation, and, where necessary, revision by the authors to uphold the principles of accuracy, originality, and academic rigor. Notably, the LLM was not utilized for the generation of novel intellectual contributions, experimental frameworks, data analyses, or conclusions, which were exclusively derived from human expertise. This judicious application of LLMs adheres to established ethical standards for AI-assisted academic writing, balancing the enhancement of textual quality with a commitment to transparency and scholarly integrity.

