# OpenReview forum: "Beyond Templates: Dynamic Adaptation of Reasoning Demonstrations via Feasibility-Aware Exploration"
_ICLR.cc/2026/Conference — ICLR 2026 Conference Withdrawn Submission_

### Official Review · Reviewer_aLUd · 2025-10-31

**Soundness:** 2
**Presentation:** 3
**Contribution:** 3
**Rating:** 2
**Confidence:** 4

**Summary:**

This paper presents DART designed to help small language models learn complex reasoning skills from datasets created by much larger, expert models. The core problem is that smaller "student" models often can't follow the difficult reasoning steps of expert "teacher" models, which hurts their performance. DART solves this by not forcing the student to copy every step. Instead, it first checks if the student model is capable of performing each expert step. If a step is too hard, DART lets the student model stop imitating and explore its own, alternative reasoning path to solve the problem, with the only rule being that it must still arrive at the correct final answer. The main contribution is this flexible adaptation method, which tailors the training data to the student model's actual abilities, leading to significantly better reasoning performance and data efficiency compared to rigidly copying the expert examples.

**Strengths:**

1. Interesting idea: The paper introduces "imitation gap." It finds the exact point where a small model gets confused by an expert's complex example and then smartly lets the small model find its own, simpler way to the right answer. The "imitation gap" transforms a low-quality, harmful training example into a high-quality, customized one. Instead of just discarding a hard example entirely, DART use the example in a new way.

2. Easy to Read: The paper is well-written and uses a simple Figure 1 to explain its main idea, making it easy for everyone to understand.

3. I really like this idea to convert useless examples to something useful.

**Weaknesses:**

1. Baselines: The authors didn't compare DART to a much simpler, cheaper alternative, like just finding and removing all the hard examples at the start. For example, only use pass@8 > 1 trajectories and remove those pass@8==0. It can also easily remove those too "hard" questions. Because "medium" examples are easy to learn. The paper only compares to "raw" datasets, which only proves it works better than using the "teacher" solutions directly.

2. Evaluation datasets: it only proves the DART method works on mathematical and STEM-related problems. It fails to test the framework on other, highly relevant logical tasks like code generation. Code is a similar task where it would be easy to check the model's answer by running unit tests.

**Questions:**

1. The DART method is very time-consuming to run. Any further analysis on this cost than line 661?

2. Will this method works on "code" problems, or some other logical datasets? I will increase my score if this method can be tested on other domains.

3. Any stronger baselines you can compare to? I think the current baseline is too naive.

---

### Official Review · Reviewer_6RPQ · 2025-10-31

**Soundness:** 2
**Presentation:** 3
**Contribution:** 2
**Rating:** 2
**Confidence:** 4

**Summary:**

The paper tackles an important research question of how we can transfer strong step by step reasoning from a larger model into a smaller model. The authors argue that directly transferring knowledge from a teacher to a student model faces the distribution mismatch problem and a good idea would be to allow the student to explore from time to time and instead of forcing the teacher steps all the time, allowing students' own generation can bring this gap down. There are some caveats here and there to make sure the chain and the exploration makes sense. Results are demonstrated on the mathematical datasets using qwen models of different sizes.

**Strengths:**

- The problem of distribution mismatch between a teacher and the student models is important to address and this work is focusing on an important research question. The research idea is well presented in the paper and it is well structured and easy to understand. Although the related work section is a bit incomplete, other areas seem to be well presented.
- Results use models of different sizes to study the scaling effect of the approach and the results are presented on a long list of mathematical datasets.

**Weaknesses:**

- The experiments are quite limited with nearly all baselines missing. The dynamic data collection, which trajectories to choose and when has been studied a lot in the past. Comparison with other baselines are needed to prove the point. For example, a similar problem is tackled in the SIKeD[1] paper where the authors discuss the exact same things, use very similar models and datasets and argue that using a different reasoning chain can assist in learning better and when to use student generations vs when to use a teacher generation. GKD[2] is another paper that discusses the mismatch between the student and the teacher generations and how to counter them. These papers are missing from the experimentation and also from related work which seems quite odd. Other papers like Dagger or PRM based approaches need to be discussed.
- If the constraint that O(T_adapt) = O(T_expert) is used, I am wondering why a RL based approach was not used compared to SFT style approach? Since individual trajectories are not considered when the final answer is correct, isn't a loss only on the final answer makes more sense instead of each token loss and something like GRPO would be more useful here?
- Models are a bit old now and Qwen 3 should have been a better choice than Qwen 2.5. Similarly, datasets like GSM8K are a bit outdated now and should be replaced with some modern more difficult dataset. Moreover, the improvements are not that great and I am not sure a lot of results are even statistically significant? There's no variance to know how much to trust the results in its current form. Moreover, this must  be compared with an approach like SIKeD as a baseline.

[1] SIKeD: https://aclanthology.org/2025.findings-acl.513/

[2] GKD: https://arxiv.org/abs/2306.13649

**Questions:**

1. Comparisons with baselines are missing in the paper and can authors compare their approach with other data mixing and trajectory selection baseline like SIKeD? It would even be better if a simple Dagger style re-labelling baseline is included. Even a simpler verifier based baseline to choose some samples and reject some samples using a PRM would have also worked. The paper is lacking all the baseline.
2. I am wondering if this approach can be made iterative? Once the model is trained, it can roll trajectories again and can further be trained. Any reasons why it was not iterative in the current version and was stopped at one iteration?
3. I am wondering what happens during inference ? Given a new query, the model generates once and it has learnt to explore during the training simulation and nothing is done differently at the test time inference. Is that correct?
4. I would be interested in knowing if the results are transferable across newer models like Qwen 3 and any reason why authors used old models of Qwen 2.5 instead of the new ones?

---

### Official Review · Reviewer_afjp · 2025-11-01

**Soundness:** 2
**Presentation:** 3
**Contribution:** 2
**Rating:** 2
**Confidence:** 3

**Summary:**

The paper proposes a framework to align expert reasoning data with the limited capabilities of small language models (SLMs). The authors argue that static expert demonstrations often fail when applied to weaker models due to capacity and distribution mismatches. DART addresses this by estimating, step by step, whether the student model can feasibly imitate an expert reasoning step, and when infeasible, it allows an autonomous exploration to find alternative reasoning paths that still yield the correct outcome. This feasibility-aware and outcome-consistent adaptation is expected to improve reasoning performance.

**Strengths:**

### 1. **Insight on The Imitation Gap**

The paper identifies and empirically validates the **“Imitation Gap”** — a critical phenomenon where continued imitation of expert reasoning harms small models once their cognitive limits are exceeded. This insight explains failure cases of previous imitation-based fine-tuning and offers a theoretically grounded basis for adaptive learning.

### 2. **Clarity of Methodology**
The authors provide a clear step-by-step formalization of DART’s simulation, adaptation, and outcome-alignment stages.

**Weaknesses:**

### 1. **Unclear Reliability of the “Imitation Feasibility” Estimate**

The core mechanism — step-wise adaptability estimation via solution simulation— hinges on Monte Carlo rollouts to decide whether a model can imitate a given reasoning step. However, the paper provides no rigorous validation of this signal’s reliability or stability across datasets and model sizes. Adaptability scores might be noisy, dataset-specific, or overly sensitive to sampling temperature and rollout randomness, undermining DART’s claimed robustness.

### 2. **Questionable Assumption About Autonomous Alternate Path Search**

The paper claims that once imitation becomes infeasible, the student can autonomously explore alternative reasoning paths. Yet this assumes the student model possesses sufficient meta-reasoning ability to meaningfully generate valid alternate trajectories. In practice, for small models (0.5B–1.5B), this assumption is likely unrealistic — these models may simply generate degenerate or circular reasoning patterns, not genuine adaptations. The paper does not quantify how often such explorations succeed or analyze failure cases in depth.

### 3. **Narrow Experimental Scope and Domain Bias**

Despite claiming generality, DART is evaluated only on structured mathematical reasoning datasets (LIMO, Math-QwQ-32B, GSM8K, etc.), all with verifiable numeric answers. These tasks make outcome alignment straightforward, but the method’s efficacy for open-ended reasoning domains (commonsense, dialogue, coding, scientific QA) remains untested. Hence, it’s unclear whether DART generalizes beyond narrow mathematical reasoning — a key limitation for a claim of reasoning alignment.

**Questions:**

N/A

---

### Official Review · Reviewer_aSf9 · 2025-11-02

**Soundness:** 3
**Presentation:** 3
**Contribution:** 2
**Rating:** 4
**Confidence:** 4

**Summary:**

The paper proposes DART, a framework that adapts expert chain-of-thought trajectories to a student model by estimating step-wise imitation feasibility via simulation, truncating when an imitation gap appears, and then letting the student explore its own continuation subject to outcome consistency. This shifts supervision from rigid step matching to model-compatible, outcome-aligned traces. Empirically, DART is evaluated on math reasoning datasets and multiple small model scales, showing improvements over static fine-tuning. Contributions are: identifying limits of static demonstrations and introducing feasibility-guided selective imitation, a Monte Carlo step-feasibility estimator, and an autonomous exploration mechanism that generates outcome-consistent alternatives, with experiments indicating better data efficiency and generalization.

**Strengths:**

The paper is original in reframing process supervision as feasibility-aware selective imitation with outcome-consistent exploration, dynamically adapting expert trajectories to a student’s capacity rather than uniformly copying steps. This formulation is clearly introduced with a concrete problem setup and algorithms, and the training signal is motivated to match the student’s own history, improving robustness. Empirically, the study evaluates across multiple math-reasoning datasets and several small model scales with well specified decoding and rollout settings, which supports reproducibility and demonstrates consistent gains over static fine-tuning. The significance is strong for aligning reasoning in resource-limited models, since the approach targets distribution mismatch and shows improved generalization and data efficiency.

**Weaknesses:**

The paper’s novelty over existing selective imitation, verifier or PRM-guided process supervision, and rejection-sampling fine-tuning is unclear; a head-to-head with these and strong test-time scaling baselines would clarify incremental value. The method relies on answer-checkable math tasks and narrow datasets, so generalization to open-ended outputs is uncertain; broaden beyond math and provide failure analyses and significance tests. Ablations are missing on key knobs that drive cost and behavior, including Nsim, temperature, truncation thresholds, and exploration beam size; report sensitivity and remove reliance on gold answers where possible. The approach appears computationally heavy due to multi-rollout simulation and exploration; disclose full wall-clock and budgeted comparisons under matched decoding budgets.

**Questions:**

Can you quantify compute and wall-clock cost for solution-simulation and exploration, and report matched-budget comparisons against strong baselines such as verifier or PRM reranking and rejection-sampling fine-tuning.

---

### Note · Authors · 2025-11-16

I have read and agree with the venue's withdrawal policy on behalf of myself and my co-authors.